# Performance under pressure in skill tasks: An analysis of professional darts

**Marius Ötting**[1,2]*, **Christian Deutscher**[2], **Sandra Schneemann**[2], **Roland Langrock**[1], **Sebastian Gehrmann**[2], **Hendrik Scholten**[2]

**1** Department of Business Administration and Economics, Bielefeld University, Bielefeld, Germany,
**2** Department of Psychology and Sports Science, Bielefeld University, Bielefeld, Germany

* marius.oetting@uni-bielefeld.de

## Abstract

Understanding and predicting how individuals perform in high-pressure situations is of importance in designing and managing workplaces. We investigate performance under pressure in professional darts as a near-ideal setting with no direct interaction between players and a high number of observations per subject. Analyzing almost one year of tournament data covering 32,274 dart throws, we find no evidence in favor of either choking or excelling under pressure.

**Citation:** Ötting M, Deutscher C, Schneemann S, Langrock R, Gehrmann S, Scholten H (2020) Performance under pressure in skill tasks: An analysis of professional darts. PLoS ONE 15(2): e0228870. https://doi.org/10.1371/journal.pone.0228870

**Data Availability Statement:** All relevant data are within the manuscript and its Supporting Information files. In addition, if the article will be published, we will make the data and the

## Introduction

The effect of pressure on human performance is relevant in various areas of the society, including sports competitions [1], political crises [2], and performance-based payment in workplaces [3], to name but a few. A broad distinction differentiates between effort and skill tasks. Success in effort tasks is dependent on motivation to perform while skill task outcomes underlie precision of (often automatic) execution. For effort tasks, such as counting digits [4] or filling envelopes [5], individuals will typically respond to increased pressure (e.g. resulting from performance-related payment schemes) by investing more effort, which given the nature of such tasks will improve their performance [6; 7; 8; 9]. However, the literature on the impact of pressure on performance in skill tasks, e.g. juggling a soccer ball [10], is inconsistent and effectively divided into two different strands of research.

On the one hand, the existing literature related to potential "choking under pressure" indicates broad agreement that performance in skill tasks declines in high-pressure or decisive situations. An individual is said to be choking under pressure when their performance is worse than expected given their capabilities and past performances [11]. While there may also be random fluctuations in skill levels, choking under pressure refers to systematic suboptimal performance in high-pressure situations. The associated empirical findings—both such that are based on experimental data but also those using field data—consistently confirm a negative impact of pressure on skill tasks. On the other hand, and to some extent in contrast to the literature related to choking under pressure, the literature related to the concept of "social facilitation" refers to potential negative but also potential positive effects of (social) pressure on performance—depending on circumstances associated with the performance. The social

corresponding R-Code available in a GitHub repository. Alternatively, the data and R-Code can also be uploaded at PLOS ONE (if possible) to provide it to the readers, as we submitted data and code together with the manuscript in the submission.

**Funding:** The author(s) received no specific funding for this work.

**Competing interests:** The authors have declared that no competing interests exist.

facilitation literature explicitly incorporates characteristics of the task and individuals' level of expertise into their analyses, and generally states that the circumstances surrounding performance play an important role regarding the impact of pressure on performance. Existing contributions focusing on potential choking have largely neglected the corresponding more comprehensive picture drawn by the social facilitation literature, by simply relating performance decrements to changes in the execution of actions, or simply distraction, generated either by rewards in case of success [12; 3] or potential penalties in case of failure [13].

Our empirical investigation of individual's performance in pressure situations is based on a large data set from a skill task, namely professional darts, comprising 32,274 individual dart throws, for a comprehensive empirical test of performance under pressure. For the professional darts players analyzed in this study, playing darts is a full time job. The top players regularly earn prize money exceeding one million Euro per year. In professional darts, highly skilled players repeatedly throw at the dartboard from the exact same position effectively without any interaction between competitors, making the task highly standardized. The amount of data available on throwing performances not only allows for comprehensive inference on the existence and the magnitude of any potential effect of pressure on performance, but also enables to track the variability of the effect across players. The literature on choking would suggest that performance of professional darts players declines in high-pressure situations. However, when considering the highly standardized task to be performed and players' high level of expertise, we do not expect dart players to choke under pressure.

The paper is structured as follows: Section 2 reviews the literature on performance under pressure, and in particular details what we consider to be advantages of the darts setting with respect to investigating performance under pressure. In Section 3, we explain the rules of darts and define what constitutes pressure situations in darts. Section 4 presents the empirical approach and results.

## Performance under pressure

### Terminology

Pressure results from individuals' ambitions to perform in an optimal way in situations where high-level performance is in demand [12]. Performance under pressure could in principle go either way, i.e. high expectations towards (the own) performance could impact performance in a negative (choking) or a positive (clutch) way—or not at all. To measure the impact of pressure, performance in pressure situations is compared to performance in non-pressure situations. Choking under pressure refers specifically to a *negative* impact of high performance expectations [14; 15] while clutch performance is described as "any performance increment or superior performance that occurs under pressure circumstances" (see p. 584 in [16]).

### Potential effects of pressure

The impact of pressure on performance crucially depends on the type of task to be performed. Tasks can be such that performance is determined mostly by effort, or alternatively tasks can be such that the skill level is the key factor for success. For effort tasks, pressure situations result in increased effort and hence improved performance [17]. For skill tasks, performance has been demonstrated to be both impaired (choking) and increased (clutching) by pressure—or not affected at all. While the effect of pressure on effort tasks is obvious and well documented, in skill tasks the potential psychological factors at play are likely more complex, such that we focus on these tasks in the following.

**Choking.** Choking under pressure in skill tasks may be related to various drivers. In particular, different skills may make use of different memory functions, namely explicit and

procedural memory, respectively [18]. Explicit memory enables the intentional recollection of factual information, while procedural memory works without conscious awareness and helps at performing tasks. Two classes of attentional theories capture choking under pressure, distraction theories and explicit monitoring theories [19; 1]. Some authors argue that distraction and explicit monitoring theories are not necessarily mutually exclusive, but rather complementary (see, e.g., [20; 21]). Distraction theories claim high-pressure situations to harm performance by putting individuals' attention to task irrelevant thoughts [20; 22]. Put in a nutshell, individuals concern about two tasks at once, since the situation-related thoughts add to the task to be performed. Given the restricted working memory individuals performance declines as focus is drawn away from the main task [23].

On the other hand, self-focus or explicit monitoring theories explicitly predict that pressure increases self-consciousness to a point where it harms performance (overattention). It can cause the skilled performer to deviate from routine actions [24]. Instead, closer attention is paid to the single processes of performance and their step-by-step control. This ties in with the concept of skill acquisition: when initially learning a skill, performance is controlled consciously by explicit knowledge as actions are executed step-by-step [25]. Over time and through practice, skills become internalized and usage of conscious control decreases. Pressure can interfere with this now automated control processes of skilled performers [26]. Under pressure, actions are no longer executed automatically as attention is redirected to task execution [19]. The overall sequence of actions is broken down into step-by-step control as in early stages of learning, resulting in impaired performance [27]. Consequently, individuals consciously monitor and control a skill they would perform automatically in non-pressure situations [28; 19].

**Other potential effects.** An alternative strand of literature suggests that 'pressure' situations do not inevitably affect performance in a negative way but may also have a positive impact on task performance—or no effect at all. The corresponding notion of social facilitation is one of the oldest paradigms within experimental social psychology (see, e.g., [29; 30]): "Generally, social facilitation refers to performance enhancement and impairment effects engendered by the presence of others either as coactors or, more typically, as observers or an audience" (see p. 75 in [31]). A potential theoretical explanation for the opposing effects of audience is that social presence facilitates *dominant* behavior [29]. Dominant behavior refers to the kind of response which is more likely: correct or incorrect. In case of, e.g., simple tasks it is more likely to perform the task correctly while individuals tend to make more mistakes when executing more complex tasks [32]. Hence, whether audience facilitates [+] or impairs [−] performance depends on the type of task (simple [+] vs. complex [−]) and/or individuals' level of expertise (expert [+] vs. beginner [−]) [33]. The presence of others increases the individuals' (physiological) arousal or drive level which in turn impairs or enhances task performance, respectively [29]. A review of 12 years of research following the drive theory suggests that their propositions are still valid [30]. Nonetheless, alternatives to drive theory have evolved in the following decades. While some non-drive theories relate audience effects to self-awareness [32], others refer to (cognitive) attention focus [34]. Though experimental research uniformly confirms that social presence affects individuals' performance, it remains unclear which mechanism mainly drives behavior. As the presence of others represents a particular case of pressure, it hence seems perfectly possible that pressure *enhances* performance—depending on the type of task and the individuals' level of expertise.

## Empirical findings for performance under pressure in skill tasks

As this paper analyses performance under pressure in a sport-related skill task, this section is devoted to previous findings from sports. There are also early non-sport studies [12; 35]. Golf

putting performance is investigated in an experimental setting, suggesting performance to be worse when subjects are put under pressure [22]. However, in high-pressure situations participants who are distracted by a secondary task (counting down from 100) outperform subjects who solely concentrate on the putting task. The latter result is explained by too much focus on the task execution induced by the additional motivation to perform well in high-pressure conditions. The additional focus disturbs task execution which normally is performed automatically. There is also further evidence for diminishing golf putting performance under pressure provided by asking 108 undergraduate students with little or no golf experience to putt a golf ball as close to a target as possible [36]. Considering different kinds of intervention methods, pressure-like situations using monetary incentives are created. Results generally confirm decreasing performance for high-pressure situations. However, the authors show putting accuracy to slightly increase under pressure when subjects had made their practice putts under self-consciousness-raising conditions.

Based on the assumption that pressure increases left-hemispheric activation which in turn is related to the controlled execution of a task and thereby to performance decrements, participants of a previous study performed a sport-related motor skill task in three blocks (in soccer, tea kwon do, or badminton) [37]. While the first two trials serve as for the introduction of pressure, the third trial is performed after participants have squeezed a softball for 30 seconds. Thereby, half of the participants activated their right hemisphere by squeezing the ball in their left hand, before again performing the task under pressure. Overall, the findings indicate performance deterioration when pressure is introduced but that the activation of the right hemisphere can eliminate this effect, thus preventing choking under pressure. However, they find no evidence for increased performance under pressure.

In a further study, a throwing task had to be performed by the participants to analyze novices' performances [38]. During the experiment, the performance expectancy within the experimental group regarding the ability to perform under pressure is manipulated. The results show a significant performance increase of the experimental group when pressure is applied, while the performance of the participants in the control group does not alter before and during pressure situations.

For a hockey dribbling task with 34 experienced participants, performance is found to be worse in high-pressure situations [28]. Results further show that within high and low-pressure conditions subjects perform better when not concentrating explicitly on the task execution. By analyzing a hockey dribbling setting with experienced hockey players, additional evidence for declining performance in pressure situations is found. However, it is demonstrated that in a high-pressure priming condition, performances are equal to those in a low-pressure situation and better (thus faster) than in a high-pressure non-priming condition [39].

For basketball novices, decreasing free throw success in pressure situations is shown [28]. This result only applies to those subjects who are asked to pay close attention to the execution process during the practicing phase. Analyzing free throw performances of competitive basketball players instead of novices supports the results [40]. Thus, participants suffer a significant decrease in free throw success when performing in a high-pressure situation induced by the introduction of an audience, videotaping and offering financial rewards for improved performance.

A further study analyzes the impact of fear of negative evaluation on performance, investigating success rates of throwing a basketball from a short distance [41]. The shots are taken from five different spots which all are placed at the distance of the free throw line. The authors find decreasing performance (thus choking) only for participants who were anxious about being evaluated negatively. For other subjects no significant differences in success rates are found.

Outside of experiments, field studies take advantage of the wealth of data on actual market participants who repeatedly perform almost identical tasks but under varying degrees of pressure. Pressure in these instances is determined by factors such as the importance of the competition considered, the current score in the competition, and the time left to play in a match.

Penalty kicks in soccer are considered to be a prototype pressure situation, as they critically affect the match outcome and the expectation to score a goal is very high. In line with the hypothesis of individuals tending to choke under pressure at skill tasks, success rates of penalty kicks in professional football are found to decline with increasing importance of success, i.e. as pressure increases [42]. However, contradictory to these results, success rates in penalty shootouts are found to increase with pressure in the German cup competition confirming clutch performance [43]. In addition, several studies focus on the "last-mover disadvantage", i.e. whether teams that go first in a shootout have an advantage over the other team resulting from higher pressure from trailing [44; 45; 46]. One of these studies finds that that last-mover teams indeed suffer from this kind of pressure [45], the other studies refute this finding and speculate the contradictory results to be a consequence of data issues [44; 46]. Potential reasons for varying success in penalty shootouts between players are that players from high-status countries a) generally perform worse and b) engage more in escapist self-regulation strategies than players from low status-countries [47].

In golf, performance under pressure is analyzed for putting [48; 49]. Analyzing the impact of the current leaderboard situation on performance, the author finds that interim results are irrelevant for performance. In particular players who are in the lead or close to the lead in the final round do not perform worse than those who are further behind. Furthermore, players' performances are constant across rounds. Between-athlete comparisons may explain this finding, which is not in line with the widely accepted hypothesis of individuals choking under pressure [50]. Considering also within-golfer comparisons, such findings cannot be replicated, and corresponding studies instead do find athletes to choke under pressure [50]. Relating choking under pressure to golfers' age, an inverted U-shaped relationship on the professionals' tour with performance under pressure peaking at age 36 is shown [51]. The success rate at the final putt of a golf tournament is found to decrease as the value associated with that shot increases [52]. Finally, golfer currently with the lead are found to underperform at the end of close contests [53].

Basketball free throws constitute another scenario that is often investigated to analyze performance under pressure. Considering data from the National Basketball Association (NBA), and modelling free throw success rates as a function of the current score, players are shown to perform much worse when their team is either trailing by 1 or 2 points, or in the lead with 1 point. Attempts are more successful when the score is tied (which equals less pressure since a miss would end in an overtime and not a loss) [54]. Further evidence for choking under pressure in professional basketball is reported with performance declining with additional pressure [55]. However, the authors show performance to be unaffected by the crowd size, the tournament round, and whether or not it is a home game for the player considered. Examining the determinants of choking under pressure, overall lower free-throw success rates are found for different groups (containing females and males, and amateurs and professionals) in case of high-pressure situations [56]. Analyzing the performance of professional basketball players who had been categorized as "clutch players" by basketball experts is also part of previous research [57]. Results show that clutch players are indeed able to increase their performance (which is measured by points scored and fouls drawn) in high-pressure situations such as the final minutes of close games, while performance of other players is not affected by pressure. Therefore, results provide evidence that clutch performers actually do exist. However, the analysis further shows no differences for clutch players' field goal percentage between low-pressure

and high-pressure situations. It is also reported that professional basketball players who maintain their performance under pressure earn higher salaries [58].

While some contradictory results have been reported, overall there still seems to be fairly evidence that professional athletes do choke under pressure, at least in some scenarios.

## Task features of the darts setting

**Empirical advantages.** Despite the effort that has already gone into studying the impact of pressure of performance, we believe that the setting of professional darts is an important addition to the existing body of literature. While we do not claim the following features to be unique to darts—as they effectively also apply to bowling, archery etc.—they are important to mention as they improve the reliability of any results obtained, compared to other more complex settings which have regularly been analyzed in past research.

First, in darts, players cannot interfere the performance of the opponent directly. In order to precisely measure the impact of pressure, analyses need to focus on such performance that is not affected by others [59]. In many other settings, such as penalty kicks in soccer, opponents can impact each other's success. As a matter of fact missing a penalty shot can be caused by the kicker's or the goalkeeper's performance, respectively, or both. The individualistic nature of darts reduces variance caused by interference of opponents present in other settings.

Second, subjects in our data are highly trained in the task they perform. Such experience is obtained from training and previous competition, the latter may or may not be covered in our sample. Observing experienced professionals vastly reduces the noise to be expected for inexperienced players with large fluctuations in performance. The separation of the impact of pressure on performance is hence much clearer in professional sports settings (compared to lab experience with amateurs).

Third and closely related to the previous point, the task to be performed in a pressure situation is more or less identical to the only task the players perform throughout the contest. The only difference is given by the specific field the player attempt to hit. In comparison, penalty shots only account for a very small fraction of actions a soccer player need to perform [60]. In line with our previous argument, estimating skill levels in pressure situations requires such separation of signal and (potentially very large) noise. If pressure is closely related to the task at hand (e.g. a penalty shot) it is hard to separate between pressure generated by the task and pressure generated by the situation.

Fourth, all players in darts are repeatedly confronted with high pressure situations. For penalty kicks or free throws, team managers may rely on the same set of players when confronted with pressure situations, namely those who they have faith in to deal with the pressure or are very skilled in the specific task. Such sample selection can be detrimental to the quality of the results and occurs especially for very specific tasks.

Overall, we believe that professional darts offers a nearly optimal empirical setting to investigate the impact of pressure on performance. Players repeatedly perform highly standardized actions, with no interference by an opponent or any teammates involved, and hardly any relevant external factors.

**Characteristics of task / dart players.** As already discussed above, the social facilitation literature suggests that the circumstances surrounding performance affects the consequences of pressure. These circumstances mainly refer to the individuals' level of expertise and complexity / difficulty of the task. As our data set includes professional dart players who are highly trained in throwing darts, we observe individuals of high expertise.

Throwing darts is a skill task which requires high motor skills in order to perform well [61]. There is a high level of standardization of individual throws as well as many repetitions of

almost identical actions, performed by professionals. Even though hitting a specific slice of the dartboard requires a high precision of movements, we assume that throwing a dart at a dartboard is less complex than, e.g., shooting a penalty (soccer), throwing at a basket (basketball), or putting a ball (golf). The more the task relies on simple, well-rehearsed responses, the smaller the chances of performance decrements. Hence, we expect performance of dart players to be unaffected by pressure. In contrast to the literature related to social facilitation, the choking literature would predict that performance in darts declines as pressure increases.

## Pressure situations in darts

For readers who may be unfamiliar with the rules of darts, we here provide a short description. The dartboard consists of 20 different slices, which differ with respect to their value (ranging from 1 to 20), and the center of the board, which is composed of two fields, namely the single bull and the bullseye. Each slice is further divided into three different parts: two single, one double and one triple field. The bullseye is the double field of the single bull. Fig 1 shows the layout of a standard dartboard, highlighting the single five segment, the double and triple eight, respectively, and the single bull together with the bullseye. The inside width of the triple and double fields is 8mm, whereas the diameter of the bullseye is 12.7mm. A darts match is typically played by two players. (There are cases of team competitions in darts but these are not considered in our analysis.) Players are standing 2.37m away from the dartboard (at the "oche"), the height of which is 1.73m (from the ground to the center of the bullseye).

While there are many possible games in darts, professional darts commonly follow the *501 up* format. In order to win a corresponding match, a player must be the first to win a pre-specified number of legs (typically between 7 and 15). Both players start each leg with 501 points and the opening throws in a new leg alternate between the two players. The first player to reach exactly zero points wins the leg, with the restriction that the dart that ultimately reduces the points to zero must hit a double field. For instance, in case a player throws a dart at the single/double/triple field of segment 20, 20/40/60 points are deducted from the player's current score. If a player hits a field that reduces his score below zero it is called a bust. The player starts

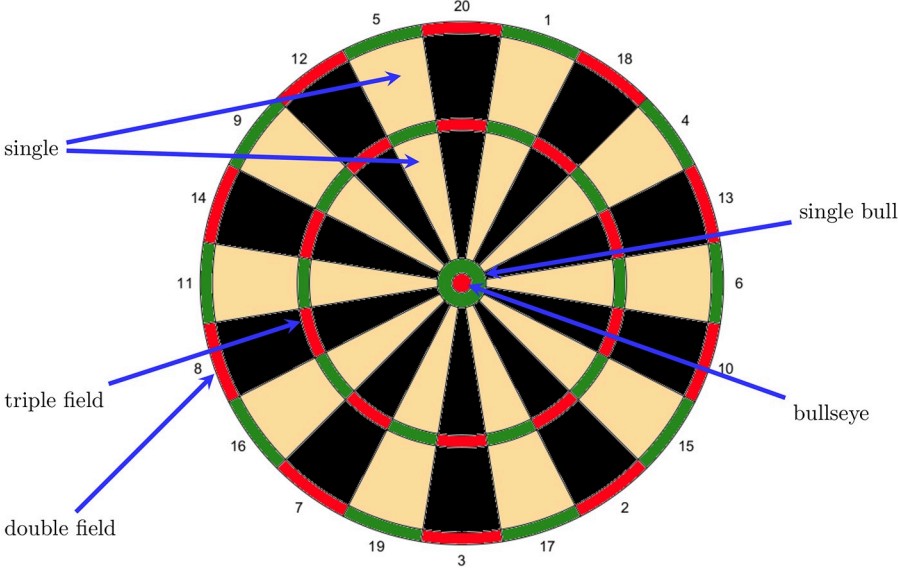

**Fig 1. Dartboard layout.**

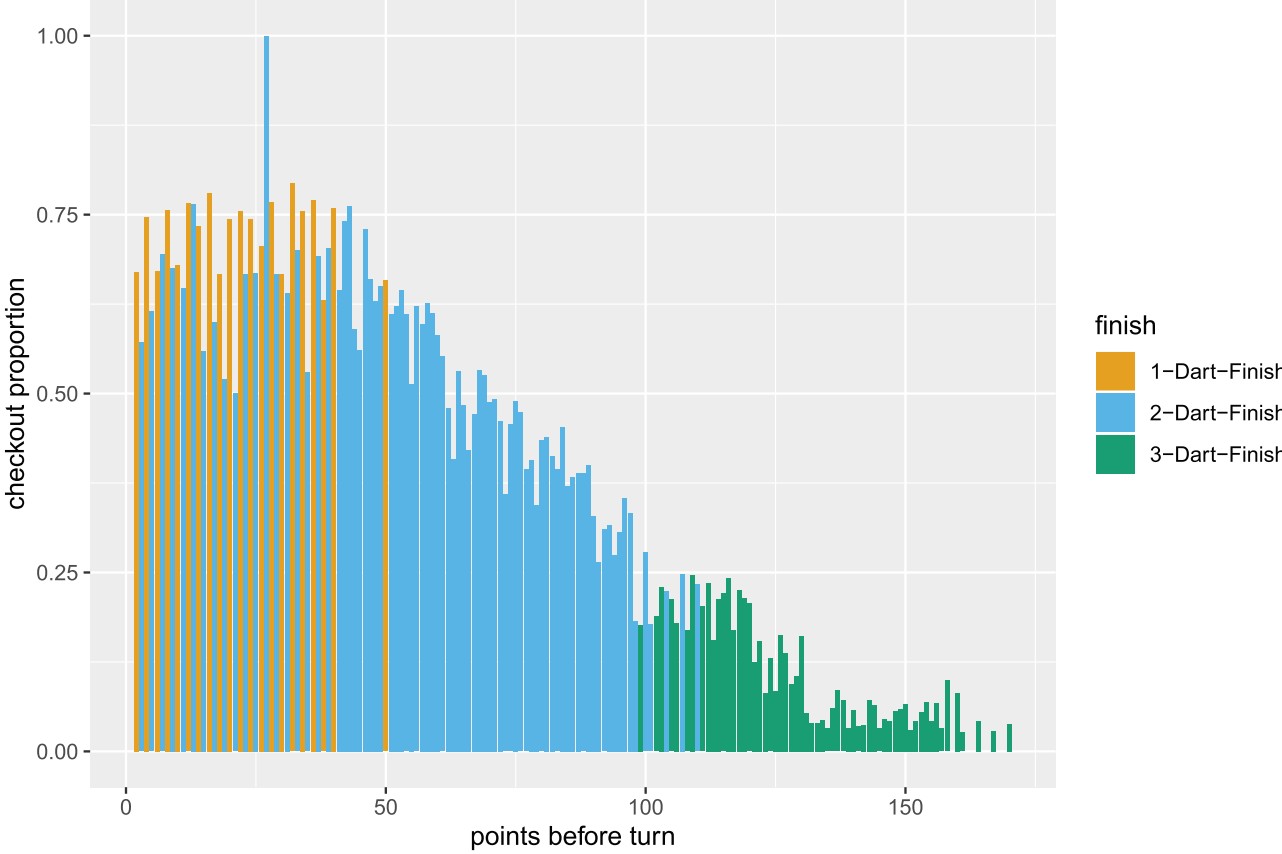

**Fig 2. Checkout proportions for the individual scores before a player's turn.** Colors indicate whether for the given score 1, 2, or 3 darts are needed for a checkout.

with the number of points he had before he busted at his following turn. The players take turns to throw three darts in quick succession. At the beginning of a leg, players consistently aim at high numbers—usually triple 20 or triple 19—to quickly reduce their points. The maximum score per dart is 60 (triple 20) and hence 180 for a set of three darts.

Once a player has the possibility to finish a leg (i.e. reach exactly zero points) with three darts (or less) during his turn, he is in the *finish region*. If he takes the opportunity and finishes the leg, this is called a *checkout*. As the last single dart has to hit a double field, the highest possible checkout is 170: two darts at triple 20 (2 × 60 = 120) followed by a dart into the bullseye (50 points). The highest checkout not requiring a bullseye is 160 (two triple 20 followed by a double 20). For some scores below 170 there are multiple combinations for a checkout while there are none for others (e.g. 159 points as there is no three darts combination that leads to exactly zero points with the last dart hitting a double field. 159 points could be reduced to exactly zero points with three darts if the last dart does not need to hit a double field, e.g. by triple 20—triple 20—triple 13. However, since all tournaments in our data are played as "double out", 159 points can not be reduced to zero within a players' turn).

We determine the likelihood of a player checking out for any given number of points left. To do so, we use information on all attempts for the given score to determine the success rate (see below). The checkout proportions for the individual scores are shown in Fig 2, which in addition indicates whether (at least) 1, 2, or 3 darts are needed for a checkout. It is important to note that there is a strategic element to the game, where players sometimes deliberately

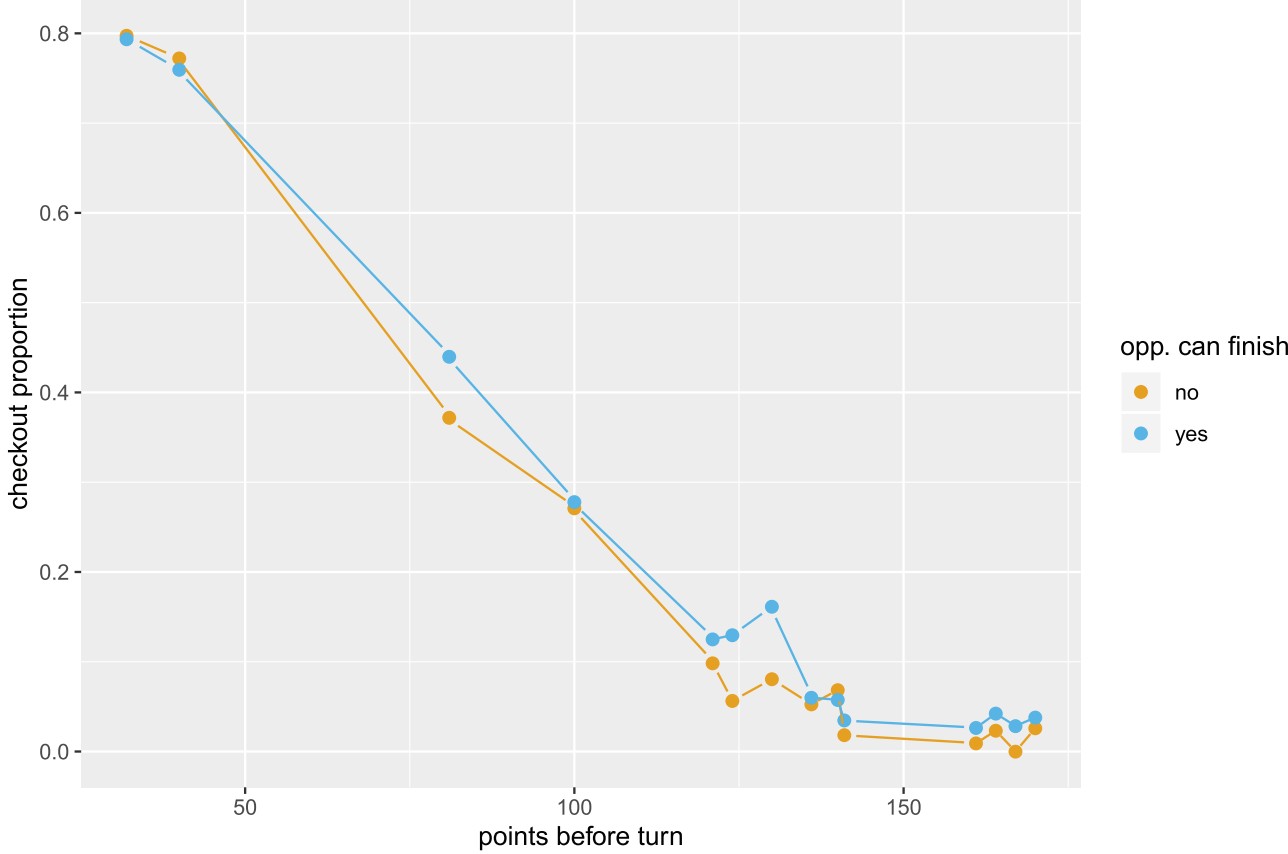

**Fig 3. Checkout proportions for situations where the opponent had a finish (blue) and those where the opponent had no finish (yellow).** Finishes with at least 100 observations in each category are shown.

attempt to set their score to a certain number for their next turn instead of checking out immediately. If, for example, Player A has a fairly high number of points to check out, say 160 points, but Player B has no finish with his next turn, then Player A could set up an easier checkout for his next turn rather than going straight for a checkout. The occurrence of such strategic behavior is corroborated by Fig 3, which shows the checkout proportions in the data for those situations. For scores above 120 the checkout proportion for Player A is usually higher if Player B has a finish (compared to situations where Player B has no finish). When having a high score left to finish, players tend to set up an easier checkout if their opponents have no chance to finish in the next turn. Such strategic behavior becomes less relevant for lower scores. For scores below 50, many of which can be checked out with one dart only, such that setting up a score is less relevant, the checkout proportions do not differ substantially between situations where the opponent had a finish and those where he had no finish. We explicitly account for such strategic considerations by restricting our sample to those observations where the opponent also has a finish.

For any given turn of a player, the level of pressure is a result from the player's own likelihood of finishing within the current turn as well as that of the opponent finishing within his next turn. Respective probabilities are estimated by the corresponding empirical proportions as described above. Following the literature, the intermediate score of a match can also generate pressure. We hence also analyze a sub-sample of throws, which are performed in situations that are very crucial to the outcome of the match. More specifically, we investigate *decider legs*,

referred to as legs where both players only need one more leg to win the match. Winning such a leg hence results in winning the match, whereas losing such a leg would result in losing the match. For example, in a best-of-19 leg match, a decider match occurs when the score is tied at nine legs apiece and leg number 19 decides the winner of the match. Pressure in decider legs is thus higher.

## Empirical analysis

The data—extracted from http://live.dartsdata.com/—covers all professional darts tournaments organized by the Professional Darts Corporation (PDC) between April 2017 and September 2018. Based on the raw data it was possible to reconstruct which player makes a throw, the score before each dart, how many legs have been played in the match, which player had the first throw in any leg considered and, of course, if the player making a throw checks out. In the data we analyze, each row, i.e. observation, corresponds to a player's turn to throw (at most) three darts. From those rows, i.e. from all sets of three darts played by a player, we consider only those instances where both the player and the opponent have the chance to check out within the given and the next turn, respectively. To ensure reliable inference on player-specific effects, we further reduced the data set to consider only those players who had at least 50 attempts to check out. The final data set comprises information on the checkout performances of $m = 122$ different players, totaling to $n = 32, 274$ observations (checkout yes/no).

## Descriptive statistics

Our response variable *Checkout* indicates whether a player managed to check out (coded as "Checkout = 1") or not ("Checkout = 0"). As detailed above, we measure the degree of pressure on a player by differentiating between his and the opponents' chances to finish a leg prior to his turn. The chance of a player checking out is quantified by the checkout proportions of all finishes from the player's current score (*CheckoutProportion*). For the opponent, the corresponding covariate *CheckoutProportionOpp* indicates the checkout proportion of the opponent's current score. In an alternative model specification, we replaced the *CheckoutProportionOpp* variable by a dummy indicating whether or not the opponent had a chance to check out with his next attempt, restricting the sample to 1-dart finishes for comparable checkout proportions. The corresponding results (not shown) were consistent with the ones presented here. To account for the ex-ante heterogeneity of players' chances to win the match, the competitive balance (*Cb*) indicates the absolute difference in the winning probabilities. Based on betting odds taken from http://www.oddsportal.com/, and after correcting for the bookmakers' margin, *Cb* can take values between 0 and 1. High values of *Cb* imply that the match is lopsided, whereas the value 0 means that both players have equal winning probabilities. Finally, as our data contains trained athletes, we are able to further control for the experience of the athlete (*Exper*), proxied by the number of years the player belongs to a professional darts organization (British Darts Organisation or PDC).

Table 1 summarizes all covariates considered. Overall, about 42% of all checkout attempts are successful. However, the probability to successfully complete a checkout is highly dependent on the number of points required: the more points are needed, the less likely is a checkout (see Fig 2).

To investigate the impact of pressure on performance, Fig 4 shows the checkout proportions for different levels of pressure, which are indicated by the colors. Due to the potential strategic adjustments discussed above, only those observations where the opponent can also finish are included. For scores above 100, the checkout proportions seem to increase with

**Table 1. Descriptive statistics for the covariates.**

|  | obs. | mean | std. dev. | min | max |
|---|---|---|---|---|---|
| *Checkout* | 32,274 | 0.420 | – | 0 | 1 |
| *CheckoutProportion* | 32,274 | 0.419 | 0.279 | 0.027 | 1 |
| *CheckoutProportionOpp* | 32,274 | 0.486 | 0.266 | 0.027 | 1 |
| *Exper* | 32,274 | 13.15 | 7.050 | 0 | 36 |
| *Cb* | 32,274 | 0.363 | 0.228 | 0 | 0.899 |

increasing likelihood of the opponent checking out, i.e. the more a player is under pressure. For lower scores there is no such clear trend.

In addition, the pressure as indicated by decider legs is investigated in Fig 5 by comparing the empirical checkout proportions in decider vs. non-decider legs. Since in only about half of the finishes the checkout proportion is higher in decider legs, there is no clear pattern indicated by these summary statistics.

## Modelling checkout performance

The structure of the data considered is longitudinal, as we model the binary response variable $Checkout_{ij}$, indicating whether or not the $i$–th player ($i = 1, \ldots, m$) checked out ($Checkout_{ij} = 1$) on the $j$–th attempt ($j = 1, \ldots, n_i$). To cover player-specific effects, and also to account for

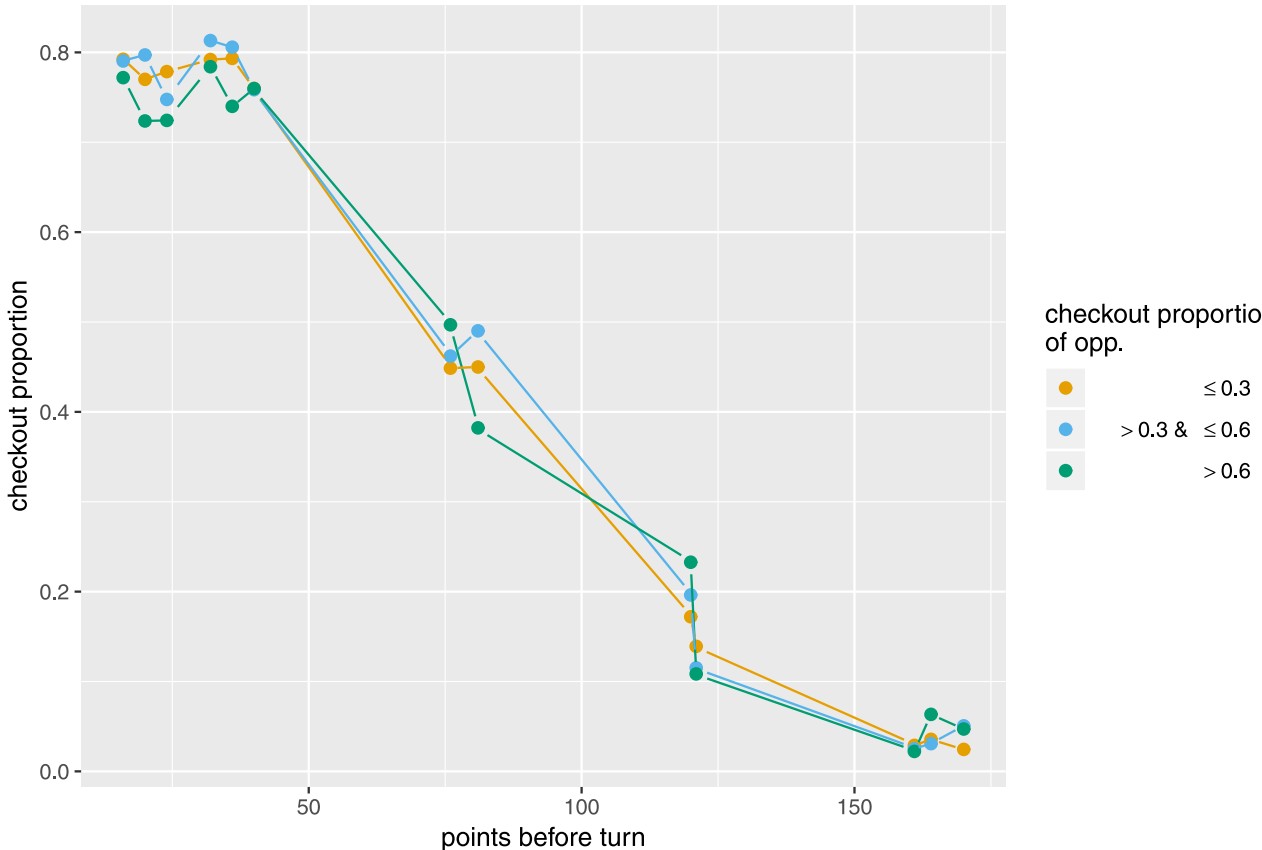

**Fig 4. Checkout proportions in pressure vs. non-pressure situation.** Specifically, checkout proportions are separated for different categories of checkout proportions *of the opponent*. Only scores with at least 100 observations per category are shown.

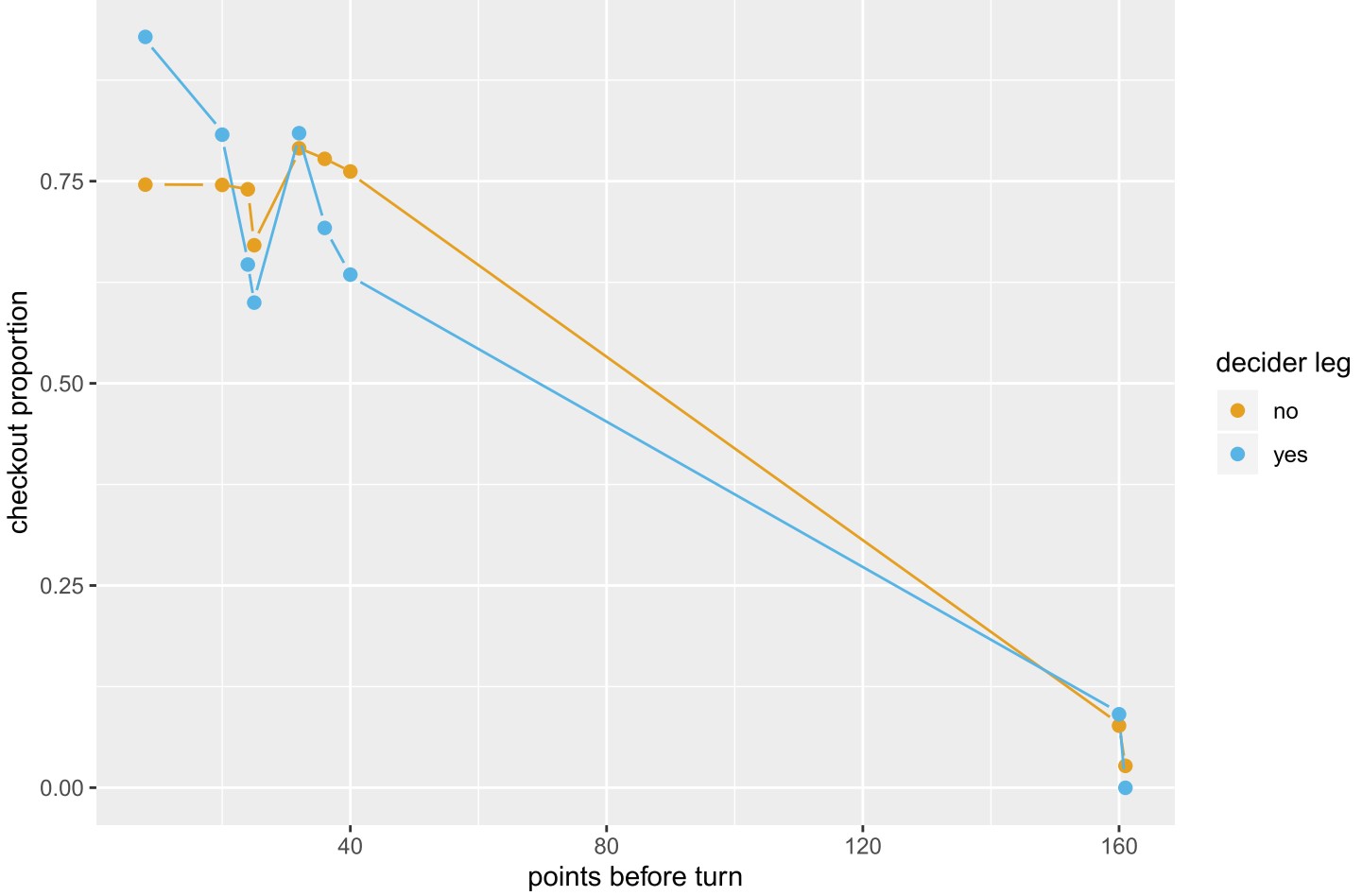

**Fig 5. Checkout proportions in pressure vs. non-pressure situation as indicated by decider legs.** Scores with at least 10 observations per category are shown.

the fact that each individual player's observations are likely to be correlated, we apply generalised linear mixed models where the linear predictor $\eta_{ij}$ contains a vector of fixed effects $\boldsymbol{\beta}$ as well as a vector of zero-mean random effects $\boldsymbol{\gamma}_i$:

$$\eta_{ij} = \boldsymbol{x}'_{ij}\boldsymbol{\beta} + \boldsymbol{u}'_{ij}\boldsymbol{\gamma}_i, \qquad i = 1, \ldots, m, \quad j = 1, \ldots, n_i,$$

with $\boldsymbol{x}_{ij} = (1, CheckoutProportion_{ij}, \ldots)'$, and $\boldsymbol{u}'_{ij}$ the subvector of $\boldsymbol{x}'_{ij}$ with those covariates for which we assume individual-specific effects. The logit function links the binary response variable, $Checkout_{ij}$, to the linear predictor:

$$\text{logit}(\Pr(Checkout_{ij} = 1|\boldsymbol{\gamma}_i)) = \eta_{ij} = \boldsymbol{x}'_{ij}\boldsymbol{\beta} + \boldsymbol{u}'_{ij}\boldsymbol{\gamma}_i.$$

The linear predictor includes all covariates considered as well as a random intercept for each player to account for player-specific effects:

$$\eta_{ij} = \beta_0 + \beta_1 CheckoutProportion_{ij} + \beta_2 CheckoutProportionOpp_{ij} + \beta_3 Exper_i + \beta_4 Cb_{ij} + \gamma_{0i}.$$

The random intercept $\gamma_{0i}$ displays the player-specific deviation from the average intercept $\beta_0$—further individual-specific effects will be considered below. These models are fitted by

**Table 2. Estimation results for the fixed effects of the turn-level model.**

| | Response variable: | | |
| --- | --- | --- | --- |
| | Checkout | | |
| | **all attempts** | **no deciders** | **deciders** |
| *CheckoutProportion* | 5.132 | 5.128 | 5.715 |
| | (0.058) | (0.058) | (0.536) |
| | p = 0.000 | p = 0.000 | p = 0.000 |
| *CheckoutProportionOpp* | 0.014 | 0.016 | −0.108 |
| | (0.053) | (0.054) | (0.423) |
| | p = 0.798 | p = 0.766 | p = 0.799 |
| *Exp* | 0.005 | 0.005 | 0.007 |
| | (0.003) | (0.003) | (0.018) |
| | p = 0.099 | p = 0.090 | p = 0.697 |
| *Cb* | −0.018 | −0.014 | 0.107 |
| | (0.068) | (0.068) | (0.545) |
| | p = 0.785 | p = 0.842 | p = 0.845 |
| *Constant* | −2.799 | −2.797 | −3.084 |
| | (0.062) | (0.062) | (0.456) |
| | p = 0.000 | p = 0.000 | p = 0.000 |
| Observations | 32,274 | 31,715 | 559 |

maximum likelihood estimation using the package `lme4` in R [62; 63]. Table 2 displays the results for the corresponding fixed effects.

The estimated coefficients associated with *CheckoutProportionOpp* are of main interest here as they display the impact of the opponent's chance of checking out during his next attempt on the player's chance to check out during his current attempt. To identify different levels of pressure connected to the intermediate score of the game, we fitted the model to different samples, distinguishing non-decider legs and decider legs. Perhaps somewhat surprisingly, evaluating the effect of *CheckoutProportionOpp* across the first two models, the more pressure a player is exposed to, i.e. the more likely the checkout of the opponent, the higher is the increase in the corresponding odds for a checkout. However, the corresponding effects are not statistically significant. For the third model, the effect is also statistically insignificant. Hence, pressure apparently does not impact performance. This is also supported by a different model formulation where we pooled all attempts and introduced a dummy variable indicating if the throw occurred in a decider leg. The corresponding coefficient is insignificant, again providing no evidence for an effect of pressure on performance (results not shown). The player-specific random intercepts $\hat{\gamma}_{0i}$, i.e. the player-specific deviations from the intercept $\hat{\beta}_0$, range (on the logistic scale) from −0.217 to 0.398.

To conduct a more fine-grained analysis of the throwing performance, we ran a second analysis in which we changed the sampling unit to single throws instead of a complete turn of three throws. When analyzing single throws instead of turns in darts, additional strategic adjustments have to be considered. If players can reduce their score to 0 with a single dart (e.g. if their score is 32), players often throw a "marker dart" with their first dart of a turn just outside of the board, such that the second dart is aimed at the marker and may be deflected into the target. To again account for such strategic adjustments, we only consider the third dart of a turn, since no marker darts are thrown with the third throw. The covariate *CheckoutProportion* is then built from the score-specific checkout proportion of the third dart of a turn. The results of fitting the model to data of single throws are shown in Table 3. As was done also for the

**Table 3. Estimation results for the fixed effects of the model fitted to data of single throws.**

| | Response variable: | | |
| | Checkout | | |
| | all attempts | no deciders | deciders |
|---|---|---|---|
| *CheckoutProportion* | 4.534 | 4.562 | 2.749 |
| | (0.308) | (0.310) | (2.586) |
| | p = 0.000 | p = 0.000 | p = 0.288 |
| *CheckoutProportionOpp* | 0.076 | 0.084 | −0.327 |
| | (0.066) | (0.067) | (0.508) |
| | p = 0.253 | p = 0.208 | p = 0.520 |
| *Exp* | 0.005 | 0.006 | −0.001 |
| | (0.003) | (0.003) | (0.021) |
| | p = 0.035 | p = 0.032 | p = 0.962 |
| *Cb* | 0.150 | 0.148 | 0.710 |
| | (0.082) | (0.082) | (0.658) |
| | p = 0.068 | p = 0.074 | p = 0.281 |
| *Constant* | −2.394 | −2.408 | −1.570 |
| | (0.129) | (0.130) | (1.041) |
| | p = 0.000 | p = 0.000 | p = 0.132 |
| Observations | 14,849 | 14,590 | 259 |

previous analysis based on turns (see Table 2), we fitted the model to data of all attempts, to non-decider legs, and to decider legs separately. The results again indicate that pressure does not impact performance in professional darts.

Since in the current model formulation we only allow for player heterogeneity in the baseline throwing performance, we further consider an extension where potential additional variation in the performance-in-pressure situations across players is investigated. The corresponding (and again insignificant) results are presented in the appendix.

## Discussion

We find no evidence that professional darts players are impacted by (high) pressure situations. While player-specific effects for performance under pressure indicate that some professional players in our sample may improve, and some may worsen their performance in pressure situations, the average effect over all players is not statistically significant. Hence, our results do not corroborate studies supporting the choking hypothesis which states that overall performance in skill tasks decreases with increasing pressure.

The difference between our findings and previous studies on performance under pressure may partly be due to the fact that in our study we consider very highly skilled individuals who have to deal with the considered type of pressure situations on a regular basis. Professional darts players are at the very top of their profession and cannot fluke out of pressure situations, which is possible in team settings where tasks can be assigned to different team members. In fact, darts players face pressure situations on a regular basis and hence gain experience in dealing with these. While throwing darts is the one skill required in the setting considered, in other professions the set of tasks is much more diverse, often combing the requirement of both, skill and effort.

The literature on social facilitation offers a possible explanation for the absence of any choking effect. Social facilitation suggests that the type of task and level of expertise greatly affect the consequences of audiences or general pressure. As all players in our data set are professionals, pressure situations should affect performance positively. However, we find positive effects

only for some but not all players. Accordingly, results cannot be attributed to the type of task. Since all of the players are of high expertise and execute the same task, the type of task should have the same effect on all players. On the one hand, "ceiling effects by performing a well-learned task" (see p. 75 in [31]) may lead to such insignificant performance effects. Hence, future research on darts players should also observe less experienced subjects in order to circumvent such ceiling effects. On the other hand, players may differ with respect to personal variables, such as self-confidence. Thus, pressure may affect performance differently depending on personal attributes. Further research on performance under pressure would benefit from including more information on personal characteristics.

Investigating semi-professional players (such as youth players) may further be beneficial with respect to a potential selection bias. Our sample may to some extent be the result of selection effects of subjects who can withstand pressure and become professionals, such that only those individuals who do not choke in pressure situations succeeded in the profession at hand and made it to the top (and hence into our sample).

The importance of coping with pressure situations has been investigated by in a qualitative study by interviewing ten international top athletes [64]. In this study, several attributes are stated as important factors for being "mental tough", such as to be in control under pressure. In a further study, again several former Olympic or world championship winning athletes are interviewed as well as sport psychologists and coaches, finding that mentally tough athletes can not only cope with pressure situations, but even use it to raise their performance [65]. An explanation for this is that individuals are either entering a "competition state" or a "threat state" when forced to pressure situations, where the former helps their performance and the latter does not [66]. Thus, to not choke under pressure is not a conscious decision but rather a state of mind which is reached subconsciously.

Throwing darts arguably is a very specific task, much less complex than other actions required to perform in under pressure situations. Our finding of individuals not choking under pressure may be due to this specific task feature. Thus, future research on performance under pressure should include characteristics of the task and individuals into their considerations as these drive pressure effects. While the setting itself would be ideal to test gender differences in performance under pressure in a specific task, women's darts does not offer the data necessary to draw comparisons. Empirical comparisons in line with the research by [67] are thus not possible at this time. Given the high number of observations for each player, further research could tackle the question if there is a memory for choking under pressure. More precisely, one could determine if choking under pressure impacts future choking under pressure, similar to a hot hand phenomenon particularly concerning pressure situations [68; 69].

Even though the social facilitation literature helps to understand the inconsistent impact of pressure on individuals' behavior, it may be the case that pressure resulting from, e.g., competing for large monetary rewards or championship titles differs from pressure due to the presence of others. Whether individuals react to pressure with enhanced or impaired performance may hence also depend on the kind of pressure they experience while performing a certain task. It would be interesting to test whether dart players react differently to pressure situations (due to interim results) when playing before an audience or no spectators, respectively. However, this scenario would only be testable in laboratory settings as there are no contests taking place without spectators.

## Appendix

In the appendix, we present the analysis of potential additional variation in the performance in pressure situations across players. This is investigated by analyzing throwing performance

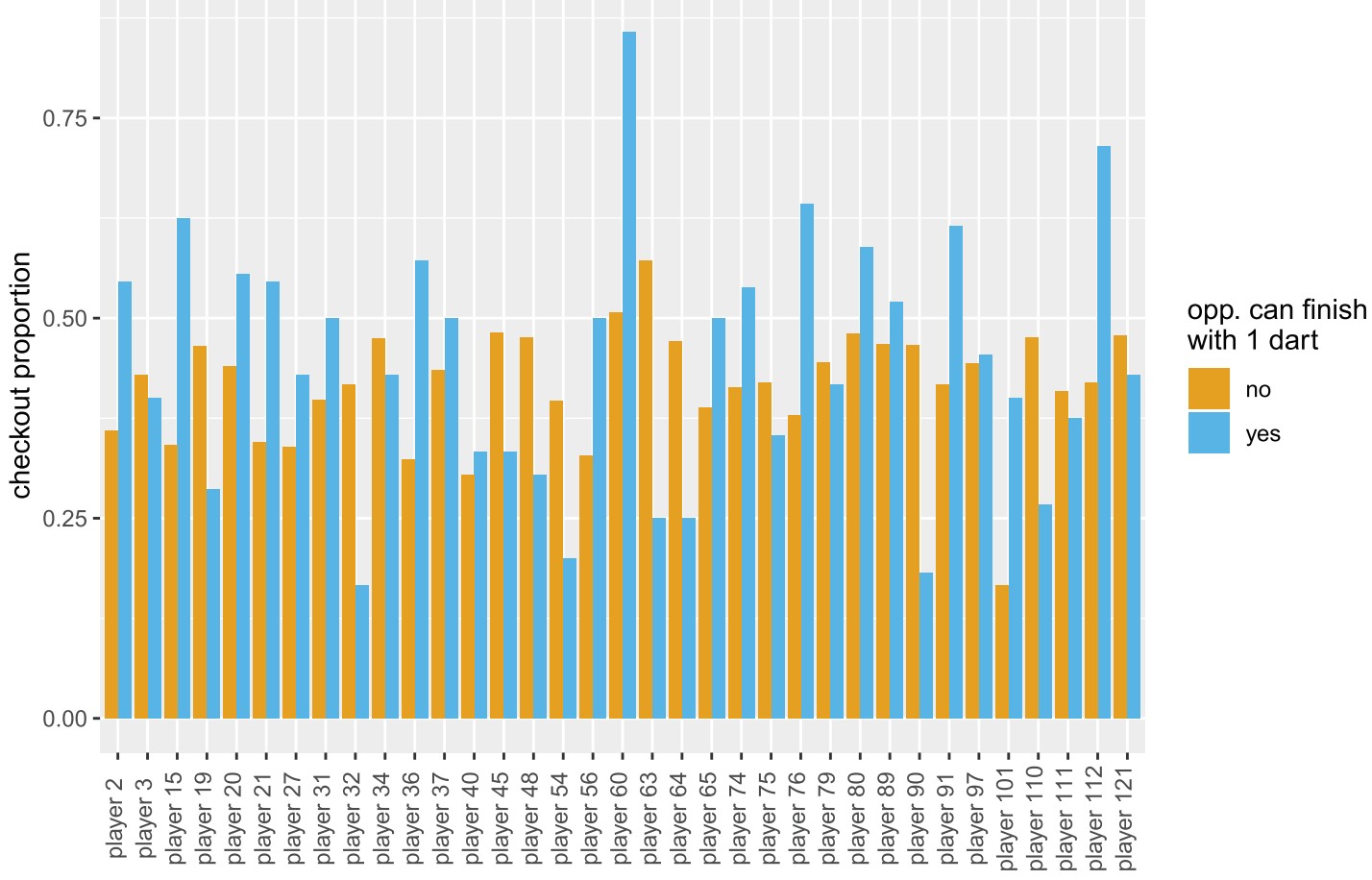

**Fig 6. Checkout proportions for situations with 2, 8, 16, 22, 32 or 36 points to checkout before the third throw of a turn.** Colours indicate whether the opponent also had 2, 8, 16, 22, 32 or 36 points left. Checkout proportions are shown for players with at least 10 observations in the corresponding subsample, i.e. third throws of non-decider legs with 2, 8, 16, 22, 32 or 36 points left.

based on individual throws. While the model presented here provides some insights regarding player-specific performances under pressure, it should be noted that it does not yield an improvement in the AIC compared to the individual-throw model considered above. To analyze scores which are of about the same difficulty, we consider the scores 2, 8, 16, 22, 32 and 36. The corresponding checkout proportion of these scores with the third dart of a turn vary between 0.408 and 0.476. The checkout proportion for all scores which can be finished with a single dart vary between 0.231 (34 points) and 0.476 (2 points). To make the throws comparable, we restrict our analysis to the above mentioned scores with checkout proportion of at least 0.4. Considering these finishes for third throws where the opponent also had a finish accounts for $n = 4,773$ single dart throws. A first comparison of the performance under pressure situation between players is investigated in Fig 6. The colors indicate whether the opponent also has a remaining score of 2, 8, 16, 22, 32 or 36, thus indicating pressure situations for the player (denoted by *OppCanFinish* below). Remarkably, there are substantial differences between the players. To extend the model formulation considered above, we include additional zero-mean random effects, $\gamma_{1i}$, which represent the player-specific deviations from the fixed effect of

**Table 4. Results of the individual-throw model with random slopes.**

| | Response variable: | | |
| | Checkout | | |
| | all attempts | no deciders | deciders |
|---|---|---|---|
| *OppCanFinish* | 0.025 | 0.037 | −1.244 |
| | (0.092) | (0.092) | (1.161) |
| *Exper* | 0.002 | 0.002 | −0.034 |
| | (0.004) | (0.004) | (0.032) |
| *Cb* | 0.426 | 0.432 | 0.440 |
| | (0.126) | (0.127) | (1.156) |
| *Constant* | −0.473 | −0.488 | 0.202 |
| | (0.074) | (0.075) | (0.553) |
| Observations | 4,773 | 4,698 | 75 |

**Table 5. Estimated fixed effects of *OppCanFinish* with the added corresponding random slope.**

| | $\hat{\beta}_2 + \hat{\gamma}_{1i}$ |
|---|---|
| Player with biggest performance improvement | 0.161 |
| Player with 2nd biggest performance improvement | 0.147 |
| Player with 3rd biggest performance improvement | 0.136 |
| $\vdots$ | $\vdots$ |
| Player with 3rd biggest performance decline | -0.128 |
| Player with 2nd biggest performance decline | -0.135 |
| Player with biggest performance decline | -0.139 |

*OppCanFinish*, leading to the following linear predictor:

$$\eta_{ij} = \beta_0 + \beta_1 OppCanFinish_{ij} + \beta_2 Exper_i + \beta_3 Cb_{ij} +$$

$$\gamma_{0i} + \gamma_{1i} OppCanFinish_{ij}.$$

As was done also for the previous analyses (see Tables 2 and 3), we fitted the model to data of all attempts, to non-decider legs, and to decider legs separately. The estimated fixed effects are displayed in Table 4. The particular pressure situation defined above, as indicated by *Opp-CanFinish*, i.e. the situations where the opponent also has 2, 8, 16, 22, 32 or 36 points left, does not have a statistically significant effect on the checkout performance. The estimated random effects $\hat{\gamma}_{1i}$ are further investigated in Table 5, displaying the sum of the estimated fixed effect of *CheckoutProportionOpp*, $\hat{\beta}_2$, and the corresponding player-specific random effect $\hat{\gamma}_{1i}$. As already indicated by Fig 6, the checkout performance in pressure situations varies substantially between players, but the model fit is not improved compared to the models presented above without additional random effects for the performance under pressure.

## Author Contributions

**Conceptualization:** Marius Ötting, Christian Deutscher, Sandra Schneemann.

**Data curation:** Marius Ötting, Sebastian Gehrmann.

**Methodology:** Marius Ötting.

**Project administration:** Christian Deutscher.

**Software:** Marius Ötting.

**Supervision:** Christian Deutscher.

**Visualization:** Marius Ötting.

**Writing – original draft:** Marius Ötting, Christian Deutscher, Sandra Schneemann, Roland Langrock, Hendrik Scholten.

**Writing – review & editing:** Marius Ötting, Christian Deutscher, Sandra Schneemann, Roland Langrock.

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
