## [Decision Letter · Decision Letter 0]

20 Sep 2019

PONE-D-19-24035

Very Highly Skilled Individuals Do Not Choke Under Pressure: Evidence from Professional Darts

PLOS ONE

Dear Mr. Ötting,

Thank you for submitting your manuscript to PLOS ONE. After careful consideration, we feel that it has merit but does not fully meet PLOS ONE’s publication criteria as it currently stands. Therefore, we invite you to submit a revised version of the manuscript that addresses the points raised during the review process.

Below you will find three reviews from scholars with expertise on the topic of your manuscript.  As you will see, two of the reviewers noted that they reviewed a previous version of this manuscript for a different journal and the manuscript has not been revised in response to those comments.  I will allow you the opportunity to revise and resubmit your manuscript if you feel you can adequately address the reviewers' concerns.  However, please note that this practice of ignoring reviewers' comments after receiving a reject decision and simply submitting the unedited manuscript to a different journal is frowned upon by many in the field.  Please try to be very responsive to each comment made by the reviewers.  If you choose to submit a revision I will send the manuscript out for re-review to the original reviewers.

We would appreciate receiving your revised manuscript by Nov 04 2019 11:59PM. To enhance the reproducibility of your results, we recommend that if applicable you deposit your laboratory protocols in protocols.io, where a protocol can be assigned its own identifier (DOI) such that it can be cited independently in the future. For instructions see: http://journals.plos.org/plosone/s/submission-guidelines#loc-laboratory-protocols

We look forward to receiving your revised manuscript.

Kind regards,

Darrell A. Worthy, Ph.D

Academic Editor

PLOS ONE

Journal Requirements:

Reviewers' comments:

Reviewer's Responses to Questions

**Comments to the Author**

1. Is the manuscript technically sound, and do the data support the conclusions?

Reviewer #1: Partly

Reviewer #2: Partly

Reviewer #3: Partly

2. Has the statistical analysis been performed appropriately and rigorously? 

Reviewer #1: Yes

Reviewer #2: No

Reviewer #3: Yes

3. Have the authors made all data underlying the findings in their manuscript fully available?

Reviewer #1: Yes

Reviewer #2: Yes

Reviewer #3: Yes

4. Is the manuscript presented in an intelligible fashion and written in standard English?

Reviewer #1: Yes

Reviewer #2: Yes

Reviewer #3: Yes

5. Review Comments to the Author

Reviewer #1: Note: I previously reviewed this article for a different journal and its content is unchanged from the submission I previously reviewed. Thus, I am providing the exact same review.

This paper has a very interesting topic and a great data set to explore psychological issues of performance. As laid out by the authors, darts appears to be an ideal setting to measure the performance of individuals. The writing and placement of their contribution within the literature is on point. However, I also have several questions and suggestions as it relates to the data and analysis. I hope my comments are helpful.

Major Comments:

1. Wow, this is really great data! With that being said, the detailed data could allow for a more precise examination of performance than the 1,2,3 dart checkout situations. I agree with the analysis done and its result, but want to know more about the likelihood of checking out from certain scores. I am a little concerned that the 1,2,3 dart finish situations are too broad of a categorization for the difficulty to checkout. From my understanding of darts (playing and watching very little), along with the authors explanation of the rules and situation (great job there), maybe all 2-dart finish situations are not created equal, it could be that 56 is much easier to checkout from than 80. In addition to table 4, I would like to know the probability of checking out from each specific score at the start of a player’s throw of 3 darts. A figure could easily display this information. Also, are there point totals from which a player’s attempt to checkout happen more or less often? I would suspect there are common point totals which players start their checkout attempts from. To address this last point, the authors could add a histogram of the scores at the start of a players turn when they have a chance to check out. In general, my concern is that there is some clumping of scores at a certain spot which make it much easier or harder for a player to checkout and the broad categorization used does not appropriately capture the difficulty.

2. Another issue closely related to #1, is the mention of strategy in not closing out in footnote 6, this brings up some questions as it relates to the data and the strategy of not checking out. At the top of page 14, the description of the data mentions the researchers know “the score before each dart”. All of the analysis in the paper is from the perspective of knowing the score before throwing all 3 darts when it is a player’s turn. Footnote 6 gets more at this idea of having data on each single dart throw as it discusses excluded throws where “the player did not try to check out with his last dart”. Since there is data on each dart throw, then why would you not look at specific dart throws where the player is on an even number and they need to throw the last dart of their turn into a specific double number? You still have the information on the probability for your opponent to check out, and you can more precisely measure performance in this moment of pressure. The 1,2,3 dart checkout scenarios could have many instances of strategies not to checkout when your opponent can’t checkout on their next turn, or situations where there are mistakes made by the player which causes them to land on an odd number and have no ability to check out on their second or last dart.

3. More specifically on footnote 6, it mentions situations when a player has a 1-dart finish, meaning that they have an even numbered score of 50 or less to start their turn. Well, they have 3 chances to make a double, and the footnote is correct that players often times may not go for the checkout on the first dart and instead get their score to an easier position such that a double 8, 16, or 20 would give them a checkout. However, the example in the footnote only discusses two dart throws, when the player would actually have three darts throws in a turn, this makes the example a little confusing. Is the exclusion of 11 observations from the case when a player has a 1-dart finish and decides not to try and checkout on the last dart throw? First, I can see why there would only be 11 excluded in this case as it means that the player had to make a mistake on their two previous throws and then be on a number likely to bust or an odd number for them to decide to not checkout. The main point here, can’t this same type of strategy happen on 2 and 3 dart finish turns? In general, strategy to not checkout may be driving the results in Table 5 as a 3-dart checkout is very difficult, players will often want to land on a good even number (e.g. 40, 32, 16) when their opponent can’t checkout on the next turn.

4. In general, to keep the current analysis in the paper, there needs to be much more discussion of player strategy to checkout and convince the reader that not checking out is a failure in performance and not a strategy decision. A quick google search returns websites with discussions on strategies to closeout (e.g. http://www.sentex.net/~pmartin/patdarts/strategy.htm), meaning the objective can often be to setup an easier closeout next turn when you are in an advantageous position against an opponent.

5. Checking out is an integral part of the game, as mentioned in the paper, matches are won when a player wins somewhere between 7 and 15 legs. This means you will need to checkout at least 7 times in a match to win. I would think checking out in the first leg of a match would have a different level of pressure than checking out for a 7th leg win to take down the match. In this sense, the legs in a match have different levels of importance and pressure as the match progresses. This seems very similar to making putts early vs. late in a round or sinking three throws early in a basketball game vs. the final minute in a close game. Taking the analysis towards a comparison of checking out in pressure vs non-pressure would more closely mirror the situations in other sports research and allow for a result that could be easily compared to other studies. If this type of analysis led to a different result than golf or three throws, it would be very compelling.

6. Section 5 which includes Tables 8 and 9 is a nice addition to tackle the issue of prize money. However, the discussion is incomplete, one sentence on the results from Table 9 in the last paragraph of section 5 is not nearly enough. Also, the discussion of expected results when adding these control variables states, “One would expect a higher Cb to be associated with lower incentives to perform well, hence a negative impact on the probability to check out”. Table 9 results indicate a positive coefficient for the Cb variable, which I understand to mean higher Cb means more likely to checkout, the opposite of the pervious statement. Maybe I am misunderstanding this, but either way this needs some clarification.

Minor Comments:

1. There are several typos, (e.g. ‘in’ instead of ‘is’, ‘price’ instead of ‘prize’).

2. There is a recent paper that could be added to the discussion of golf and performance under pressure (Hickman, Kerr and Metz 2019). The study shows players perform better in 2nd vs 1st. This may fit the story in this paper of performing best in 2 and 3-dart checkout situations (slightly behind) when the opponent has a 1-dart checkout (slightly ahead) on their next turn.

3. My understanding of darts is that there are some nuances about having a number very close to zero and busting (going past zero) which likely deserves some discussion in the paper. There could be a situation where a 1-dart checkout where a single throw will either win you the leg or have you bust; this could be a throw with enormous pressure compared to other situations.

4. While Table 7 is being used to try and indicate better performance under pressure, this result of all 122 players checking out more often when their opponent could checkout next turn vs. not checkout leads me to the strategy conclusion discussed in my previous comments. If all players are checking out less in the “no pressure” situation, then maybe the lower rate of checking out is a strategy decision for the end of leg play.

Reviewer #2: The manuscript depicts results from a large trial field study of dart throwing performance. The authors’ analysis results suggests that among dart throwers performance may improve under pressure. They see this conclusion as interesting because it is at variance with the usual finding from such studies showing that no there is no performance increase under pressure.

I was very interested in the manuscript. My reading suggested that the study was well-conducted and well-analyzed. The use of alternative models and alternative ways of approaching the data (including presenting the data from, an individual-difference perspective) is always a plus for a manuscript, in my opinion. My opinion is that the results are interesting enough to be publishable.

However, I did not think that manuscript itself optimally packaged these findings.

One major source of my concern was rooted in my perception that the manuscript did not fairly reflect the “lay of the land” with respect to the possibility that performance can increase under pressure.

For example, in attempting to describe when performance might increase under pressure, the authors make a broad distinction between effort tasks and skill tasks, and use that as a justification for the study (p. 2). The manuscript is written to imply that darts is more of an effort task, but that status seems be assumed. If my perception is correct, one concern is that the authors do not present any independent evidence validating that assumption.

Moreover, the authors do not link to an especially relevant literature: social facilitation/social impairment. The fact that this literature is often overlooked by researchers who have focused on the absence of evidence for “clutch” performances (e.g., performance enhancement under pressure) has long mystified me, and given the ghosting of this literature, it is not surprising that the authors of this manuscript have also overlooked it.

In my view, that is an important oversight. That social facilitation/social impairment literature is quite relevant to the issue treated in this manuscript. That literature suggests that arousal (which may be conceptually similar to “pressure”) might improve performances on tasks that require the emission of “dominant responses.” Roughly, these are responses that are well-learned in a given context.

In fact, the demonstration of social facilitation effects has been confirmed by meta-analyses (see Bond and Titus, 1983), and has been linked to physiological, cognitive, affective, and self-presentational mechanisms. For examples, see:

1. Blascovich, J., Mendes, W. B., Hunter, S., & Salomon, K. (1999). Social facilitation as challenge and threat. Journal of Personality and Social Psychology, 77, 68-77.

2. Bond, C. F., & Titus, L. J. (1983). Social facilitation: A meta-analysis of 241 studies. Psychological Bulletin, 94, 265-292.

3. Harkins, S. G. (1987). Social loafing and social facilitation. Journal of Experimental Social Psychology, 23, 1-18.

4. Huguet, P., Galvaing, M. P., Monteil, J. M., & Dumas, F. (1999). Social presence effects in the Stroop task: Further evidence for an attentional view of social facilitation. Journal of Personality and Social Psychology, 77,1011-1025.

This body of research is important to the present manuscript because it clearly points to three things: (a) task performance CAN improve under pressure; (b) the idea that whether one gets performance impairment or performance improvement may be moderated by the type of task to be performed, and (c) there are several kinds of psychological variables that may be linked to performance improvements.

Indeed, I raise this literature because it contradicts the authors’ assertion (on p. 2) that “However, there is broad agreement in the literature that performance in skill tasks, e.g. juggling a soccer ball (Ali, 2011), declines in high-pressure or decisive situations.” Prompted by my knowledge of the above literature, my immediate thought was “Oh yeah? WHICH literature?” It is certainly the case that, as is well-documented by the authors, both laboratory studies of sports-like performances and most analyses of real-world sports performances supports the conclusion that players are not “clutch” performers in that they do not improve performances under pressure. However, as I have noted above, there IS ALREADY A LITERATURE suggesting that for some kinds of tasks (maybe dart-throwing is one of them) and for some kinds of people (perhaps dart-throwing experts are those kinds of people), performances MAY improve under pressure (e.g., in high arousal states).

One of my obvious recommendations, then, is that because it suggests that performance improvement under pressure may sometimes occur, the authors need to touch base with the social facilitation literature.

However, it should be noted that other elements of the sports-like task performance literature has recently been more congenial to the performance improvement perspective than has, perhaps, characterized the past.

Examples (not a comprehensive review) of this literature are:

Preventing motor skill failure through hemisphere-specific priming: Cases from choking under pressure.

Beckmann, Jurgen; Gropel, Peter; Ehrlenspiel, Felix.

Journal of Experimental Psychology: General. Vol.142(3), 2013, pp. 679-691.

(see results of Experiment 3).

Choking vs. clutch performance: A study of sport performance under pressure.

Otten, Mark.

Journal of Sport & Exercise Psychology. Vol.31(5), 2009, pp. 583-601.

Do clutch players win the game? Testing the validity of the clutch player's reputation in basketball.

Solomonov, Yosef; Avugos, Simcha; Bar-Eli, Michael.

Psychology of Sport and Exercise. Vol.16(Part 3), 2015, pp. 130-138.

Changes in putting kinematics associated with choking and excelling under pressure.

Gray, Rob; Allsop, Jonathan; Williams, Sarah E.

International Journal of Sport Psychology. Vol.44(4), 2013, pp. 387-407.

Pitching and clutch hitting in Major League Baseball: What 109 years of statistics reveal.

Otten, Mark P; Barrett, Matthew E.

Psychology of Sport and Exercise. Vol.14(4), 2013, pp. 531-537.

Choking and excelling at the free throw line. [References].

Worthy, Darrell A; Markman, Arthur B; Maddox, W. Todd.

The International Journal of Creativity & Problem Solving. Vol.19(1), 2009, pp. 53-58.

Enhanced expectancies improve performance under pressure. [References].

McKay, Brad; Lewthwaite, Rebecca; Wulf, Gabriele.

Frontiers in Psychology. Vol.3 2012, ArtID 8.

These citations support the notion that even in the “sports performance-like” literature, there have already been reports (especially in recent years) of increases in performance under pressure, as well as articulations of possible mechanisms by which such performances increase. I obviously would recommend that this manuscript touch base with this literature, and as a consequence, tone down (a little) the implication in their manuscript that their data are totally discrepant with the existing literature.

This is especially important when the authors write: “Hence, following the literature, performance in darts would be expected to decline as pressure increases. Against the current state of research on motor tasks, we find (nearly all) professional darts players to excel in pressure situations. We argue that highly skilled individuals are able to digest pressure situations towards a positive outcome, while those who choke under pressure decide towards other professions. This emphasizes the importance of possession of skills for the impact of pressure on performance.” As illustrated by the references I have provided the literature is NOT uniform in supporting choking under pressure.

In this regard, then, I would recommend that the authors add a definition of the term “clutch” to their terminology section, that they review some of the studies suggesting that clutch performances can occur, that they link to some of the conditions under which clutch performances can occur, and that they describe some of the mechanisms that may produce clutch performances. This can all be done, of course, with the caveat that in the literature choking performances tend to be much better documented and seem to be more common than clutch performances.

Note that, in my view, my “packaging concern” does not diminish the interest value of the authors’ findings. Only via the publication of these kinds of data will one gain insight into (what I believe) are the critical questions: (1) on what kinds of tasks will performance increase under pressure and on what kinds of tasks will it decrease, and (2) what are important psychological factors that are at play in such changes (e.g., expertise, attention, etc.).

The authors seem to define choking under pressure as “specifically to a negative impact of high performance expectations (Baumeister and Showers, 1986; Hill et al., 2009).” That definition may be a little restrictive. Choking may also be viewed through a motivational lens in that it may require the desire to do one’s best.

I understand that section 2.4 is intended to try to provide a context for why dart throwing might be a particularly good context to investigate how performances might change under pressure. I agree that the sport of dart throwing has advantages (relatively standardized conditions, outcomes that are not influenced by opponent) that might serve to quiet random variance. However, a counterargument lies in convergent validity: when an outcome (e.g., choking under pressure) is repeatedly observed across very different contexts, one gains extra confidence in the outcome because the confounding variables that are unique to a given context are often not present in other contests. Hence, with convergence it is increasingly unlikely that variables unique to a context caused the result. Thus, although soccer penalty kick outcomes are directly influenced an opponent, golf tournament performances are not. Similarly, while laboratory studies of choking often use non-expert participants, field studies often show the same choking phenomenon in experts. The authors imply that dart thrower expertise is crucial to the clutch performance data that they report (e.g., “The stark difference between our findings and previous studies may partly be due to the fact that in our study we consider very highly skilled individuals who have to deal with 21 the considered type of pressure situations on a regular basis.”). Given results from other field research employing experts, this seems to be an unlikely explanation. To be more specific, as implied by the social facilitation literature, expertise is NOT the only variable that may be important to clutch performances. Studies of many sports (e.g., golf; baseball) have shown that even expert players sometimes choke, or they sometimes do not exhibit either choking or clutch in their performances.

I had concerns about the notion of “pressure” derived from Table 1 and Table 2. I understand that Table 1 is presented to provide a basis for the pressure labels presented in Table 2. However, there is no justification for the Table 2 categorical labels applied to the conjoint probabilities (no, low, moderate, high) depicted in Table 1. The authors’ labels suggest that their view of pressure is determined by how easy both the player’s task is (making one dart is easier than making three darts) and how easy the opponent’s task is. However, this system is not well articulated in the manuscript, and no external validation evidence for the system is presented. Extrapolating from psychophysics research, one needs external validation to determine how much felt pressure changes as the win probability changes, and whether and how felt pressure is related to both the difficulty of one’s own task as well as the difficulty of the opponent’s task. Might it be the case that the two variables do not contribute equally to felt pressure? For example, I can easily see that my task (1 vs. 2 vs. 3 darts needed to win) might contribute more to felt pressure than my knowledge of my opponent’s task.

Moreover, why must these probabilities be translated into categories? Could they not be used as a continuous predictor as a proxy indicator of pressure

In addition, one cannot always expect the change in felt pressure to be linearly related to the change in probability across the probability scale. That is, a change in win probability from.95 to .85 may not produce the same change in felt pressure as a change in win probability from .15 to .05.

I don’t know if these worries are enough to derail the manuscript, but I certainly do think that a thorough discussion of links between pressure and win probability, and the assumptions that the authors made in translating win probability data into pressure categories, is needed.

Given where it is placed, the inclusion of ideas from contest theory (p. 20) seems like an afterthought. If contest theory makes a priori predictions about the patters pf data to be expected, those predictions seem to be better placed in the introduction near to where the effort vs. skill distinction is first raised. Note that this effort vs. skill distinction is not the same as presented in the social facilitation/social inhibition literature. The literature comparable to the “effort” literature would probably lie in the social loafing domain of social psychology. However, I wonder about the relevance of the effort idea. Effort has not generally been posited as an explanation for choking (given the assumption that people choke, even when they are trying to do the best that they can). Hence, I doubted whether this effort-related material added much to the manuscript.

One other very minor point is that I do not know if Frontiers requires articles to be formatted in compliance with the guidelines set forth in the APA publication manual. If that is the cased, then this manuscript exhibits frequent noncompliance. A partial list of noncompliant items includes missing page headers, missing figure captions, misformatting table captions, and the misformatting of references (e.g., issue numbers are no reported only in special cases; no colons before page numbers; use the ampersand instead of the word “and”). If compliance to APA style is required, the authors clearly need to sit down with the manual and alter their manuscript to comply with the manual’s guidelines.

Reviewer #3: I have attached my report, and it can also be found below. Please note that I also served as a reviewer for a submission of this paper to another journal, and as the submission was not changed across these reviews I did not update my report.

Summary: The authors employ throw-level data from professional darts tournaments over the course of one year to examine whether choking under pressure occurs in darts. Higher-pressure situations in this context are defined as those where a player’s opponent is likely to finish on their next turn, conditional on the player himself being within three darts of checking out, compared to situations where a player’s opponent is unlikely to finish. Choking, meanwhile, is defined as a decrease in the likelihood of checking out. By these definitions, the authors find that darts players actually perform better under pressure, i.e. that they are not susceptible to choking under pressure.

Comments:

I had two primary concerns with this paper, both of which are related to how choking under pressure in conceptualized in this context:

1. As noted in my summary above, the authors defined a high-pressure situation as one in which the win probability was lower (because the opponent was doing better), conditional on having a shot to win the leg, or checkout, that turn. However, I am concerned about the aptness of this definition, on which the paper’s conclusion of course relies. While it is difficult to provide an objective definition for a “high-pressure” situation bar, e.g., capturing biometric measures of stress, I would not think that pressure would be increasing in the opponent player’s chance of winning, conditional on having the opportunity to check out yourself. Consider, for instance, a case where a player has a 50% chance of winning compared to a case where a player has a 1% chance of winning; I would say that the first case is higher-pressure because “there’s more on the line to lose” (this could be framed in the context of expectations-based reference dependence, where the loss in utility following a lost opportunity to checkout is not as substantial when a player has a 1% chance of winning because their reference point was that they would fail to do so). In fact, defining a high-pressure situation as one in which a game is “close” is an established approach in the literature (Cao et al, 2011; Toma, 2017). Yet this is different from how the authors define choking in this paper, where the lower the chance a player has of winning, conditional on having a chance that turn, the higher the pressure, even when this means that the game is actually less “close” as in the example above. Since the evidence of choking under pressure presented here is of course contingent on the definition of a high-pressure situation, this naturally substantially affects my perspective on how the results ought to be interpreted.

2. A crucial assumption the authors must make in order to identify choking under pressure in this context is that all else is held constant when comparing play across situations in which the opponent is one dart, two darts, three darts, or more away from checking out. However, my understanding is that players may adjust their strategy according to their opponent’s position. For instance, if their opponent has no chance of checking out during their next turn, a player, especially one who needs either two or three throws to finish, might play more conservatively, throwing in a way that somewhat decreases their chance of checking out this round but positions them better for the following turn. This interpretation is exactly consistent with the results: When a player needs either two or three throws and their opponent cannot checkout their next turn, they are significantly less likely to checkout on the turn in question compared to a case where it is possible for their opponent to checkout; this difference is no longer significant when the player needs just one dart to checkout and thus conceivably has a more narrow strategy set. Thus, the higher probability of checking out on a given turn when the opponent has the chance to check out on their next turn seems quite conceivably due to the differences in optimal strategy across these cases rather than improved performance under pressure.

The ideal analysis would control for these differential strategies across cases (and this is why free throw or soccer penalty kicks are an appealing context in which to study choking – there is just one optimal strategy regardless of the position in the game: try to make the shot/goal). One possible approach would be to compare performance under these different scenarios (the darts required for self vs opponent) when the overall match is close vs not. If we observed that players were more likely to checkout when the overall match was close, conditional on the position they were in in a given lag, this would seem to provide more convincing evidence that darts players indeed excel under pressure. Footnote 7 suggests that no significant results were found using this analysis (which I would think should be highlighted!), though it was unclear whether this specific test was employed.

In addition to these two main concerns, noted below are strengths of the paper as well as some additional concerns and suggestions:

1. I appreciate that the authors took seriously the psychology literature that differentiates, for instance, the theoretical predictions of the impact of choking in effort- vs skill-driven tasks – this is an important literature to understand when exploring performance under pressure and the authors did this justice in the literature review. One suggestion, however, would be to avoid generalizing the findings in this paper to behavior in the labor market in the abstract and conclusion, as this is exactly an instance of crossing from a skill-driven to a (typically in the labor market) effort-driven task, and, as noted, the theory of choking under pressure would predict very different types of behavior across these domains.

2. The paper includes a section on why other studies may not have correctly identified choking under pressure. I indeed believe that this is an important section to include in order to credibly assert that the novel findings in this paper may be true across the board, but I had concerns about a number of the claims made:

a. The authors argued that it’s difficult to disentangle pressure from “nuisance situations” such as weather conditions or the quality of the course. However, I see no reason to believe such nuisance conditions would be correlated with high-pressure situations, and so this should not pose a challenge to the identification of choking under pressure. Similarly, they argue that the small number of e.g. penalty kicks in soccer render the data sufficiently noisy as to make choking difficult to tease apart – I would see this as a specific concern if player fixed effects cannot be netted out (as in Dohmen, 2008) such that you may just have stronger players kicking in high-pressure situations, but otherwise a small number of observations alone should not invalidate results that have been found to be significant.

b. The authors note that fatigue may enter in as high-pressure situations are those in which players are more likely to be fatigued because they have been playing longer or in a more intense game. Toma (2017), however, addresses this using free-throw data by showing (1) that choking only occurs in the final 30 seconds of a close basketball game and not before, when players are presumably similarly fatigued, and (2) that the number of seconds a player has played in a game does not predict choking.

c. The authors argue that coaches may tend to put players best able to handle pressure in high-pressure situations, but this would seemingly work against the finding that players choke under pressure.

d. The remaining points are specific only to certain types of analyses (e.g. soccer or experimental data) and thus cannot explain why findings across the board may not be valid.

3. These data seem like an interesting opportunity to look at heterogeneities in choking across players (once the right definition of choking is nailed down!), and while I appreciate that the authors took a step in this direction I think there’s more that could be done with the analysis:

a. First, just a somewhat picky note that the authors’ claim is not seemingly supported by their evidence given that they do not have any data suggesting others may choke under pressure: “We argue that highly skilled individuals are able to digest pressure situations towards a positive outcome, while those who choke under pressure decide towards other professions.”

b. I like the idea behind Figure 2, but I think I would have found more informative a graph with actual rather than predicted checkout probabilities across opponent scores for players with sufficient data; that way, not only the intercept would change, but we would actually see differences in choking susceptibility (this is seemingly a motive underlying Model 2, which I appreciate!). Perhaps in a new version of Figure 2 bins of different player “types,” by choking propensity, could be created and graphed.

4. The authors work with a great, fine-grained dataset, which is exactly what would be required for this type of analysis!

Some minor additional comments:

1. It took some time for the difference between Table 1 and Table 4 to sink in for me - My interpretation of the difference is that Table 1 looks at the probability of winning the leg overall while Table 4 looks at the probability of checking out that turn. This would be helpful to clarify, especially as the logic behind why the two are differentially relevant for the analysis was not made clear (this gets back to my second point – presumably the probability of winning a leg can’t be considered as a main outcome because this is dependent on opponent play, but even the strategy for the current turn is influenced by the opponent’s standing, so this still seems like a concern when considering checking out for a given turn).

2. For more visually-oriented folks, it may be helpful to plot the degree to which the changes in probabilities of checkouts correspond with the defined pressure in the situation; in other words, create a figure combining the information in Table 1 and Table 2 such that pressure is on the x axis and the probability of checking out is on the y axis.

3. I think there is no question that darts is a skill- rather than effort-driven task, so I did not find the discussion in Section 5 to be necessary. That said, some of the variables included in this analysis, e.g. the prize amount, would seemingly be strong proxies for the pressure in a situation (the higher the prize amount, the higher the pressure) and so could be incorporated in this way. The null effects of the controls alone suggest to me that, consistent with the null effects in the choking literature often found for e.g. more important tournament games, players may internalize more the pressure of making vs breaking a play at a given moment rather than any larger stakes of an overall game that wouldn’t be localized to a particular moment.

4. The statistical analysis tests for significant differences in checking out across opponent position only compared to the reference of the opponent not having the opportunity to check out the next turn; it would be useful to test for significant differences across the different opponent positions as well.

6. PLOS authors have the option to publish the peer review history of their article (what does this mean?). If published, this will include your full peer review and any attached files.

Reviewer #1: No

Reviewer #2: Yes: JOHN J SKOWRONSKI

Reviewer #3: Yes: Mattie Toma

---

## [Author Response · Author response to Decision Letter 0]

5 Nov 2019

The response to the reviewer comments are included in an attached file.

---

## [Decision Letter · Decision Letter 1]

28 Nov 2019

PONE-D-19-24035R1

Performance under pressure in skill tasks: An analysis of professional darts

PLOS ONE

Dear Mr. Ötting,

Thank you for submitting your manuscript to PLOS ONE. After careful consideration, we feel that it has merit but does not fully meet PLOS ONE’s publication criteria as it currently stands. Therefore, we invite you to submit a revised version of the manuscript that addresses the points raised during the review process.

Two of the reviewers from the first submission evaluated your revision.  Although, they noted some improvements, it appears that there is still work to be done in order to make this manuscript suitable for publication.  Please try to address their comments in a revision.  I will likely send the revised manuscript back to these same reviewers and ask them to evaluate whether all of their comments have been addressed.  I hope to make a final decision on publication in the next round, so please do your best to address the comments made by the reviewers.    

We would appreciate receiving your revised manuscript by Jan 12 2020 11:59PM. To enhance the reproducibility of your results, we recommend that if applicable you deposit your laboratory protocols in protocols.io, where a protocol can be assigned its own identifier (DOI) such that it can be cited independently in the future. For instructions see: http://journals.plos.org/plosone/s/submission-guidelines#loc-laboratory-protocols

We look forward to receiving your revised manuscript.

Kind regards,

Darrell A. Worthy, Ph.D

Academic Editor

PLOS ONE

Reviewers' comments:

Reviewer's Responses to Questions

**Comments to the Author**

1. If the authors have adequately addressed your comments raised in a previous round of review and you feel that this manuscript is now acceptable for publication, you may indicate that here to bypass the “Comments to the Author” section, enter your conflict of interest statement in the “Confidential to Editor” section, and submit your "Accept" recommendation.

Reviewer #1: (No Response)

Reviewer #2: (No Response)

2. Is the manuscript technically sound, and do the data support the conclusions?

Reviewer #1: Yes

Reviewer #2: Partly

3. Has the statistical analysis been performed appropriately and rigorously? 

Reviewer #1: Yes

Reviewer #2: Yes

4. Have the authors made all data underlying the findings in their manuscript fully available?

Reviewer #1: Yes

Reviewer #2: Yes

5. Is the manuscript presented in an intelligible fashion and written in standard English?

Reviewer #1: Yes

Reviewer #2: Yes

6. Review Comments to the Author

Reviewer #1: The approach to examining the data in a more detailed manner is a great improvement. However, I have a few comments and suggestions.

1. The figures in the revised paper are quite cluttered and it is difficult to see exactly what is going on with so many data points and lines. Page 9 of the revision states “the more points are needed, the less likely is a checkout (see Figure 2).” The second figure in the response to reviewers is a viewer friendly manner in which to display this information than the current Figure 2 in the revised paper. While the revision removes analysis using the number of darts to finish, I think a discussion of the 1,2, and 3 dart finish could be helped by using the second figure in the response to reviewers. Along this same general thought, the first figure from the response to reviewers does an excellent job of showing there are a handful of very common point totals from which players start their turn with an opportunity to check out. The first figure in the response to reviewers could be included to help the discussion of strategy and common starting points on pages 7 and 8 of the revision. What if the figures you currently have in the revised paper (Figures 2 thru 4) only displayed point totals to checkout with more than 500 or 1,000 observations? This could help with clarity as the figures are currently quite messy with all the points. This limitation would leave anywhere from 10 to 20 data points, making it visually easier for the reader.

2. As mentioned in the response to reviewers, the first figure with the frequency of different point totals to checkout would seemingly indicate certain point totals for 1-dart finishes are extremely popular and thus there is likely a strategic element to start the next turn with those exact points. The authors take this to mean, lets exclude all the checkout attempts when the opponent can’t checkout on their next turn from the regression. As I might argue, strategy doesn’t exist when we have a 1 dart finish starting at a popular number. For example, 40, which would be double 20 to checkout, has an enormous amount of checkout opportunities. Could you not look at how players perform when an opponent has chance to checkout next turn vs. not at this common number? This information is buried there in Figure 2, but I think there is still an argument to include the inability of the opponent to check out as a part of the regression. What needs to be made sure of is that the player is not choosing to strategically avoid going for the check out. To me, an opponent with a chance to closeout when I am at 40 would represent pressure and opponent without a chance to closeout would be no pressure. Of course, there are varying probabilities for your opponent to checkout, and your regressions are capturing how this may impact the player, but you are also removing many instances when the player is unlikely to be making a strategy choice to not checkout. This comment is a very lengthy way to say, restrict the sample to 1-dart finish situations for the player and report those results as well, but include in the regression throws when the opponent cant checkout on the next turn as well.

3. The “pressure” variable in the revision is the probability of your opponent checking out, but doesn’t pressure depend on a player’s chance to win or lose the leg, the difference in probability to checkout between the players. As the analysis is currently setup, one interprets the CheckoutProportionOpp as the impact the opponent’s chance to check out has on the probability of the player checking out, holding constant the players chance to checkout. This is possibly quite different from a story where the leg is a close competition and I feel pressure, the current “pressure” variable in the revision could include many scenarios when the leg is not a close competition. I suggest using the difference in checkout probability as a measure of pressure, with the argument that greater pressure exists when the difference is very small and less pressure exists when the difference is large.

4. Table 1 in the revision indicates that CheckoutProportion ranges from 0.027 to 1, the second figure in the response to reviewers indicates the probability to checkout from any single number to be at a maximum of approximately 0.8. Is this difference a mistake or is there a part of the sample restriction that makes the probability of 1 occur? You could remove observations where the likelihood of checking out is extremely high (e.g. 1) or extremely low.

Reviewer #2: P1-D19-24035-R1

Performance under pressure in skill tasks: An analysis of professional darts

This is my second viewing of this manuscript. The authors received a considerable amount of feedback on their first draft, and they obviously worked very hard to incorporate that feedback into their revision. The authors’ responsiveness is to be commended. More importantly, in my view the authors have produced a new manuscript that, in my opinion, is better than the first.

It is good enough to rise above the publication threshold? My reading of this document prompted me to have some concerns that would give me some hesitation about seeing this version published.

One concern that I had was expository. While making a substantial effort to incorporate reviewer feedback into their new draft, the authors never quite seemed to be able to abandon some of the approaches now conflict with the new revisions. Thus, the manuscript was not presented in a manner that was as coherent and internally consistent as one might like. If a revision is again called for, I would ask the authors to go through their document carefully to ensure consistency across the document.

For example, in the current draft the authors continue to expound on the “advantages” of the darts setting. This includes that (1) performance is not directly influenced by others, (2) performers are highly trained; (3) task to be performed in a pressure situation is more or less identical to the only task the players perform throughout the contest; and (4) all players in darts are repeatedly confronted with high-pressure situations. In what way(s) are these advantages? One way in which these are an advantage may be statistical: The relatively individualistic nature of the task may work to quell stray variance in analyses (e.g., caused by the interference form the performances of others). However, the rest of the features? These characteristics are certainly not unique to darts: Many such as bowling would seem to present at least one of the characteristics. Indeed, many sports (e.g., bowling) may present them all. Moreover, the fact that darts players are highly trained, while true, is relatively meaningless for those tests of choking that have found evidence of choking in tasks where everyone in a highly trained expert (e.g., top-level professional sports).

Moreover, I do not think that in making this claim the authors have quite abstracted one lesson of the social facilitation/social impairment literature: that the characteristics of the task help to determine the outcome one sees in co-actor performance situations (which presumably are one form of pressure situations). Hence, the nature of the task that is performed will help to determine whether one observes clutchness, choking, or nothing at all in performances. Hence, from the social facilitation/social inhibition view, the things that the authors list are not “advantages” but instead are “task features” that may help to determine the outcome that one observes. Indeed, one of these features (the fact that the motor skill involved (dart throwing) is relatively simple in comparison to many other motor skills needed in sport) would lead me to the prediction that in darts tasks the phenomenon of choking under pressure should be reduced, eliminated, or even reversed (e.g., clutch).

It is this point that, for me, is where the manuscript has its interest. The “choking” literature has focused on choking, but the social facilitation/social impairment literature suggests that choking in pro sports may not be inevitable. Some circumstances may produce evidence of “clutchness,” and some might produce no difference from non-clutch situations. The task is one of the controlling variables, and the facilitation/impairment literature says that the more the task relies on simple, well-rehearsed responses, the smaller the chance of performance decreases (choking).

I would like to see the authors more explicitly and forcefully advocate this position. Why? The authors mention in their cover letter that they did not know about the social facilitation/social impairment stuff because the choking literature never cites it. Thus, in my view the authors can educate the choking researchers about the lessons learned about performance and pressure that have already been documented.

In this regard, the authors need to clean up their language a bit. At one point, they describe social facilitation as a “theory.” That’s wrong. Social facilitation/social impairments are observed performance outcomes. These outcomes supposedly reflect the impact of various arousal-prompted mechanisms (mere presence, evaluation apprehension, cognitive distraction) on tasks, with the outcomes theoretically moderated by individual expertise and task simplicity/difficulty (that’s the theory).

The authors probably over interpret their data. Sure, on a purely descriptive level, some results seem to reflect clutchness, and other results seem to reflect choking. However, in reality, in the authors’ analyses none of these effects is significant. Hence, unless results are so strong that they are “trending,” in my opinion it is probably best practice to simply characterize all the results as non-significant (and some would even use this characterization for “trending” results.

In this regard, stats mavens like me would probably like to know the p-levels of the non-significant choke/clutch tests presented in Tables 2 and 3.

Moreover, I might argue that tests of significant effects have not provided the strongest tests of clutchness/choking. Might not this optimally strong test involve an analysis of the CHANGE in the effect obtained on low-pressure vs. high-pressure situations? In this regard, then, I might have expected to see whether performance changed significantly for the checkout proportion opp variable from the no deciders trials to the deciders trials.

Though non-significant, the fact that the data pattern may have shifted from the no decider trials to the decider trials (an analysis new to this version) again illustrates an important point that I made in my earlier review – that one needs to be careful about (and maybe independently verify) the pressure that accompanies various trial types. One might, for example, expect that decider trials early in a match may not contain as much felt pressure as decider trials that occur late in a match. This idea bears some relation to the Wells. et al. golf idea that pressure is maximized in golf when a player is close to the lead in the final round of a tournament (as opposed to earlier rounds). This suggests that the authors might want to look at the performance during various decider trials in a match – early, middle, late, and final (where, conceivably, one throw can win a match).

The authors presented results across players. I have two thoughts about doing so. The first point concerns the magnitude of the coefficients for the individual players. Are any of them different enough from 0 that they fall out of the range expected from a random distribution? The point is that I wonder if it is necessary to name the players instead of labeling them with a meaningless descriptor. I know that other researchers in the area have explicitly avoided using real names to avoid the use of their results for the purpose of calling some players “chokers.”

One final point linking back to the social facilitation/social impairment issue. The application of that literature to the present task assumes that playing in front of an audience or with/against others applies “pressure”. (The darts task may be seen as an example of the kinds of competitive/social tasks that are featured in that literature). Is that assumption plausible? That is, is that social pressure different from the kind of pressure that can come from non-social sources (e.g., playing for large monetary rewards)? The mechanisms posited for the social facilitation literature would probably suggest that the answer to this is “no” , especially if one considers the mental interference produced by evaluation apprehension to be a special case of broad evaluative concerns. However, I know of colleagues who might try to make a case that there is a different kind of pressure involved if one is playing for a million dollars versus if one is playing to be acknowledged as the champion of one’s city. This latter point leads to the possibility that clutch/choking may also be related to the nature or kind of pressure that is experienced during task performance. The authors, at their discretion, may wish to raise this point in their discussion.

7. PLOS authors have the option to publish the peer review history of their article (what does this mean?). If published, this will include your full peer review and any attached files.

Reviewer #1: No

Reviewer #2: No

---

## [Author Response · Author response to Decision Letter 1]

11 Jan 2020

We again would like to thank all referees for their careful reading of our revised manuscript. We made an effort to address all comments that were made. The additional analyses as motivated by the reviewers' comments we believe have improved the level of persuasiveness of our paper, while comments concerning the interpretation of our results have improved the readability. 

We also made a number of smaller changes as requested. Detailed point-by-point responses are given in the attached response to reviewers document.

---

## [Decision Letter · Decision Letter 2]

27 Jan 2020

Performance under pressure in skill tasks: An analysis of professional darts

PONE-D-19-24035R2

Dear Dr. Ötting,

We are pleased to inform you that your manuscript has been judged scientifically suitable for publication and will be formally accepted for publication once it complies with all outstanding technical requirements.  I sent your revision back to one of the original referees and they felt that all their comments had been addressed.  Thank you for your contribution to the journal and the field, and congratulations on your latest publication.  

With kind regards,

Darrell A. Worthy, Ph.D

Academic Editor

PLOS ONE

Additional Editor Comments (optional):

Reviewers' comments:

Reviewer's Responses to Questions

**Comments to the Author**

1. If the authors have adequately addressed your comments raised in a previous round of review and you feel that this manuscript is now acceptable for publication, you may indicate that here to bypass the “Comments to the Author” section, enter your conflict of interest statement in the “Confidential to Editor” section, and submit your "Accept" recommendation.

Reviewer #1: All comments have been addressed

2. Is the manuscript technically sound, and do the data support the conclusions?

Reviewer #1: (No Response)

3. Has the statistical analysis been performed appropriately and rigorously? 

Reviewer #1: (No Response)

4. Have the authors made all data underlying the findings in their manuscript fully available?

Reviewer #1: (No Response)

5. Is the manuscript presented in an intelligible fashion and written in standard English?

Reviewer #1: (No Response)

6. Review Comments to the Author

Reviewer #1: (No Response)

7. PLOS authors have the option to publish the peer review history of their article (what does this mean?). If published, this will include your full peer review and any attached files.

Reviewer #1: No

---

## [Editor Report · Acceptance letter]

6 Feb 2020

PONE-D-19-24035R2 

Performance under pressure in skill tasks: An analysis of professional darts 

Dear Dr. Ötting:

I am pleased to inform you that your manuscript has been deemed suitable for publication in PLOS ONE. Congratulations! Your manuscript is now with our production department. 

With kind regards,

on behalf of

Dr. Darrell A. Worthy 

Academic Editor

PLOS ONE